# Bruxism associated with short sleep duration in children with autism spectrum disorder: The Japan Environment and Children's Study

**Masahiro Tsuchiya**[1]*, **Shinobu Tsuchiya**[2,3], **Haruki Momma**[4], **Ryoichi Nagatomi**[4,5], **Nobuo Yaegashi**[6,7], **Takahiro Arima**[8], **Chiharu Ota**[7], **Kaoru Igarashi**[2,3], **the Japan Environment and Children's Study Group**[¶]

1 Department of Nursing, Tohoku Fukushi University, Sendai, Miyagi, Japan, 2 Division of Craniofacial Anomalies, Tohoku University Graduate School of Dentistry, Sendai, Miyagi, Japan, 3 Department of Orthodontics and Speech Therapy for Craniofacial Anomalies, Tohoku University Hospital, Sendai, Miyagi, Japan, 4 Department of Medicine and Science in Sports and Exercise, Tohoku University Graduate School of Medicine, Sendai, Miyagi, Japan, 5 Division of Biomedical Engineering for Health & Welfare, Tohoku University Graduate School of Biomedical Engineering, Sendai, Miyagi, Japan, 6 Department of Obstetrics and Gynecology, Tohoku University Graduate School of Medicine, Sendai, Miyagi, Japan, 7 Department of Development and Environmental Medicine, Tohoku University Graduate School of Medicine, Sendai, Miyagi, Japan, 8 Department of Informative Genetics, Environment and Genome Research Center, Tohoku University Graduate School of Medicine, Sendai, Miyagi, Japan

¶ Membership of the Japan Environment and Children's Study Group is provided in the Acknowledgments.
* tsuchiya-thk@umin.ac.jp

**Data Availability Statement:** Data are unsuitable for public deposition due to ethical restrictions and legal framework of Japan. It is prohibited by the Act

## Abstract

Bruxism, the involuntary activity of masticatory muscles, is common among individuals with autism spectrum disorders (ASD). Although bruxism is bidirectionally associated with sleep issues, whether an infant's sleep duration contributes to the development of bruxism remains unknown. In this study, a dataset (n = 83,720) obtained from the Japan Environment and Children's Study, a nationwide birth cohort study, was subjected to multiple imputations using logistic regression analysis with adjustments for several maternal and child-related variables. The aim of this study was to assess whether shorter sleep duration in the neonatal period additively affected the high prevalence of parent-reported bruxism (PRB) among children with ASD. The prevalences of ASD and PRB in the participants were 1.2% and 7.2%, respectively, and the odds ratio of the increased risk of PRB prevalence in individuals with ASD (95% confidence interval) was 1.59 (1.31–1.94) after covariate adjustments. Importantly, shorter sleep duration in the neonatal period (at one month of age) was significantly associated with an increased risk of PRB prevalence in individuals with ASD. The increased occurrence of bruxism, known to be highly prevalent among children with ASD, is associated with shorter sleep duration, particularly in the neonatal stage. Based on our results, a better understanding of the development of bruxism in individuals with ASD would provide valuable information for the prevention of oral diseases.

on the Protection of Personal Information (Act No. 57 of 30 May 2003, amendment on 9 September 2015) to publicly deposit the data containing personal information. Ethical Guidelines for Medical and Health Research Involving Human Subjects enforced by the Japan Ministry of Education, Culture, Sports, Science and Technology and the Ministry of Health, Labour and Welfare also restricts the open sharing of the epidemiologic data. All inquiries about access to data should be sent to: jecs-en@nies.go.jp. The person responsible for handling enquiries sent to this e-mail address is Dr Shoji F. Nakayama, JECS Programme Office, National Institute for Environmental Studies.

**Funding:** The Japan Environment and Children's Study was funded by the Ministry of the Environment, Government of Japan. The funding source had no role in the study design, analysis and interpretation of data, the writing of the report, and in the decision to submit the article for publication.

**Competing interests:** The authors have declared that no competing interests exist.

## Introduction

Bruxism is an involuntary and repetitive oral parafunctional activity, which includes teeth clenching, grinding, bracing, and/or thrusting, specified as either sleep or awake bruxism [1, 2]. Bruxism appears with tooth eruption, tends to be habitual with growth [3], and is reportedly more prevalent in children than in adults (14–20% in children, 8% in adults <60 years old, and 3% in adults >60 years old) [4]. Habitual and/or excessive bruxism potentially aggravates orofacial tissue damage, such as tooth wear or temporomandibular disorders [5, 6]. Bruxism has multiple causal factors comprising pathophysiological and psychosocial issues [7–9]. Development of bruxism related to other underlying diseases, including neurodevelopmental disorders such as autism spectrum disorder (ASD) and cerebral palsy, is considered secondary bruxism [8]. A systematic review showed that bruxism is approximately four times more prevalent among individuals with ASD than in those with typical development [10]. In particular, since some medications for ASD potentially aggravate bruxism [7, 11], the risk of developing oral health issues such as tooth wear and orofacial pain is high in individuals with ASD [12, 13]. Thus, further research is required to understand the occurrence of bruxism in children with ASD for properly managing oral health problems.

ASD is a constellation of neurodevelopmental disorders with multiple delays and behavioral deviations that manifest during early childhood, predominantly in males, with a worldwide prevalence of approximately 1 in 100 children [14, 15]. As early diagnosis of ASD and proper intervention involving both the affected children and their parents is essential, recent advances in multidisciplinary interventional approaches in early childhood have contributed substantially to the improvement of physiological and psychosocial development [16, 17]. As bruxism's etiology [10, 18], individuals with ASD show a higher prevalence of sleep problems, with divergent patterns that emerge in early childhood [19, 20]. Appropriate sleep behaviors in early childhood contribute to healthy physical and mental development in children [21, 22]. Studies have shown a bidirectional etiological association between bruxism prevalence and sleep issues [1, 23, 24]. Our latest work using a dataset from a nationwide prospective birth cohort study also showed that shorter sleep duration in the neonatal stage additively increased the occurrence of bruxism in children [24]. Thus, sleep issues in early childhood potentially exacerbate the high prevalence of bruxism among individuals with ASD; however, information regarding the association between sleep disorders in infants who later develop ASD and the occurrence of bruxism in early childhood is lacking.

Herein, using a dataset from the Japan Environment and Children's Study (JECS), an ongoing nationwide, multicenter, prospective birth cohort study [24], we examined whether shorter sleep duration in early childhood prospectively impacts the prevalence of parent-reported bruxism (PRB) in children with ASD more than that in typically developing children.

## Methods

This cross-sectional study followed the "STROBE" guideline for cross-sectional studies [25] (S1 Text).

### Study design and participants

The JECS was conducted in accordance with the Declaration of Helsinki (announced in 1975 and revised in 2008), and the protocol was reviewed and approved by the Institutional Review Board on Epidemiological Studies of the Ministry of Environment and the Ethics Committees of all participating institutions (no. 100910001 and no. 2023–017), as described previously [26, 27]. The aim and procedure of the study were explained to all participants, and written

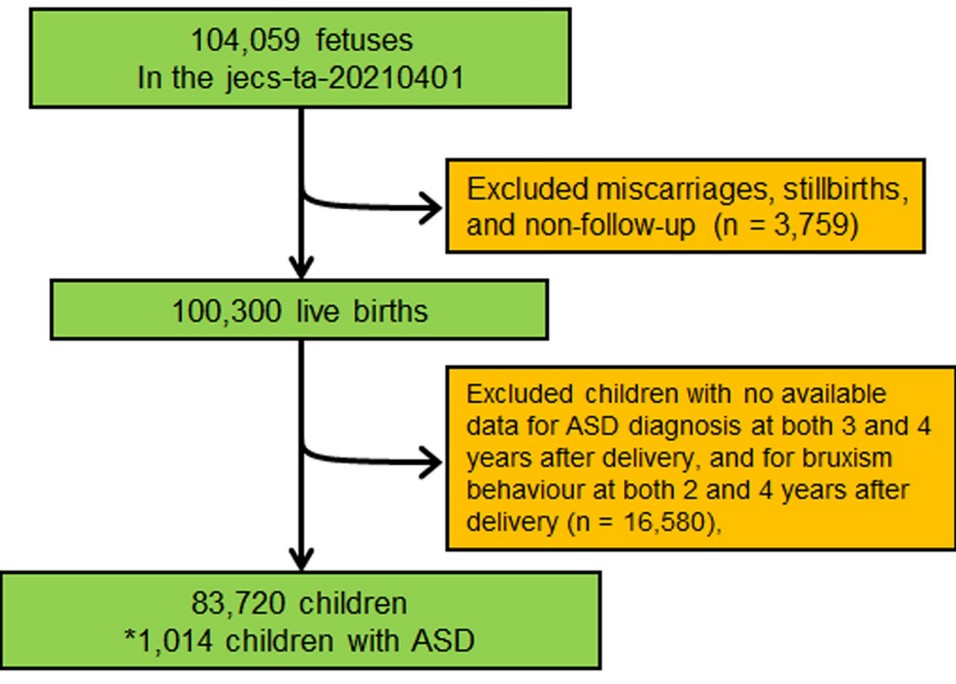

**Fig 1. Flow chart of the study participants.**

informed consent was obtained before their participation. This study was conducted as a part of the JECS and anonymized data were used. Recruitments of the JECS started from January 2011 to March 2014. This study was based on the jecs-ta-20190930-qsn and jecs-qa-20210401 datasets, which were released in October 2019 and April 2021, respectively.

Pregnant women were recruited in the JECS during their first prenatal examination by cooperating healthcare providers in local government offices between January 2011 and March 2014. After the participating mothers completed a self-administered questionnaire, medical doctors and trained nurses performed clinical measurements and summarized the medical record transcripts. Of the 104,059 pregnancies from 15 Regional Centres included in the JECS, 3,759 resulted in miscarriages, stillbirths, or loss of follow-up. Furthermore, 16,580 participating mothers did not respond to the questionnaires, which asked about ASD diagnosis at the ages of 3 and 4 years or prevalence of PRB at the ages of 2 and 4 years. After excluding these participants from the analysis, the final sample included 83,720 children (Fig 1).

## Occurrence of bruxism (outcome measure)

Prevalence of PRB was assessed in children aged between 2 and 4 years using a questionnaire reported by the caregivers, which included Yes/No questions, such as "Does your child have bruxism?" [24, 28].v PRB included all types of bruxism such as grinding and clenching, observed as either sleep or awake bruxism. As consistent prevalence of PRB would improve the reproducibility of the results obtained using parent-administered questionnaires, participants with bruxism reported by caregivers at both time points (2 and 4 years of age) were placed in the PRB group and analyzed further. Thus, prevalence of PRB was considered a binary variable defined by its absence or presence.

## Prevalence of ASD (main exposure measure)

The prevalence of ASD in participants was determined using questionnaires mailed 3 and 4 years after delivery. Parents or caregivers of the participants were asked the following question: "Has your child ever been diagnosed with the following diseases and/or disorders by a doctor?" In the section on various childhood diseases and/or disorders, there was a checkbox for "ASD". The participant was considered to have ASD if the ASD checkbox was marked at each age point. In our cross-sectional design, the prevalence of ASD was recorded as a binary variable, being either absent or present.

## Sleep duration in early childhood (secondary exposure measure)

The daily sleep duration of the participants in early childhood was estimated at 1, 6, 12, 18, and 36 months after delivery, using a follow-up questionnaire. Parents or caregivers marked the questionnaire by drawing lines through checkboxes indicating 30-min intervals from 12:00 am on each day to 12:00 am on the next day [24, 29]. Sleep duration was considered a continuous and categorical variable in the analysis. For categorizing sleep duration at each time point, the participants were categorized into four 2-h groups (e.g., each 2-h group ranges from equal to or below 13 h to more than 17 h at 1 month of age) by arranging the median in the central group at each time point. Furthermore, participants were categorized into two groups ($\leq$13 h and >13 h), which were used as the binary variable to estimate the interaction between short sleep duration at 1 month of age and ASD prevalence.

## Covariates

The detailed design of the questionnaire has been described previously [24, 26, 30]. Briefly, sociodemographic characteristics, lifestyle, and health status in mother-infant dyads, which have been previously reported as confounding variables [1, 31], were included as covariates in the analysis models. Information, such as annual household income, maternal educational level, smoking habits, and drink intake, was assessed using a self-administered questionnaire filled by the mothers during pregnancy. Maternal age at delivery and child sex were retrieved from the medical record transcripts. The presence of siblings was also assessed using a follow-up questionnaire at 4 years postpartum. All data were retrieved using medical record transcripts and self-administered questionnaires. Additionally, the prevalence of congenital anomalies was ascertained in 8,764 (8.5%) infants from medical record transcripts and questionnaires. Details regarding data processing, validation, and verification of congenital anomalies have been described previously [27, 32].

Using the data, the participants were categorized into different groups based on the following variables: annual household income in Japanese yen (<2 million, 2–4 million, 4–6 million, or $\geq$6 million); maternal educational level (junior or high school, junior college [technical or junior college], or university [university or graduate]); maternal smoking history (never smoked, stopped smoking before or during pregnancy [previously did but quit before realizing current pregnancy; previously did but quit after realizing current pregnancy], currently smoking); alcohol intake (never consumed, stopped drinking, or current drinker); sex of the child (male or female); siblings of the child (presence or absence); and the prevalence of congenital disease (presence or absence).

## Statistical analysis

Of the 83,720 participants, data on ASD diagnosis at 3 and 4 years of age were missing for 2,380 (2.8%) and 5,589 children (6.7%), respectively. Data on PRB was missing for 2,378

(2.8%) and 5,717 (6.8%) participants at 2 and 4 years of age, respectively. Multiple imputations using the multivariate normal imputation method with the "missing at random" assumption were applied to the missing data [33]. An imputation model that included all variables (including the main exposure and outcome variables) used in the main analysis was independently applied to 10 copies of the data, each of which contained suitably imputed missing values. According to Rubin's rules, the imputed values of the variables should be estimated using the means and adjusted standard errors obtained from the observed data [34].

The baseline characteristics of the patients are summarized in Table 1. Maternal age at delivery and the child's sleep duration are presented as median with interquartile range and

**Table 1. Baseline characteristics of children (n = 83,720) participating in the JECS (2011–2015).**

| | Control | | ASD | |
|---|---|---|---|---|
| Child's PRB | Absence, n (%) | Presence, n (%) | Absence, n (%) | Presence, n (%) |
| | 76,823 (92.9) | 5,884 (7.1) | 887 (87.6) | 126 (12.4) |
| *Age at delivery, median (IQR)* | | | | |
| | 31 (28, 35) | 31 (27, 35) | 32 (28, 36) | 32 (29, 35) |
| *Infant's sleep duration at months after delivery; mean (SD) in hours* | | | | |
| 1 | 14.9 (3.4) | 14.5 (3.5) | 14.4 (3.7) | 13.3 (4.6) |
| 6 | 13.6 (2.2) | 13.4 (2.3) | 13.4 (2.3) | 13.5 (1.6) |
| 12 | 12.8 (1.9) | 12.8 (2.0) | 12.8 (1.9) | 12.8 (1.8) |
| 18 | 12.3 (1.8) | 12.2 (2.0) | 12.2 (2.0) | 12.2 (1.8) |
| 36 | 11.5 (1.6) | 11.5 (1.6) | 11.4 (1.6) | 11.4 (1.1) |
| *Child's sex* | | | | |
| Male | 38,743 (92.0) | 3,370 (8.0) | 673 (87.0) | 100 (13.0) |
| Female | 38,080 (93.8) | 2,514 (6.2) | 214 (89.3) | 26 (10.7) |
| *Sibling(s)* | | | | |
| Absence | 18,543 (89.1) | 2,266 (10.9) | 296 (83.4) | 59 (16.6) |
| Presence | 58,280 (94.2) | 3,618 (5.8) | 591 (89.8) | 67 (10.2) |
| *Household income (million yen/ year)* | | | | |
| <2 | 3,917 (91.6) | 358 (8.4) | 51 (93.4) | 4 (6.6) |
| 2 to <4 | 26,124 (92.5) | 2,127 (7.5) | 320 (85.7) | 53 (14.3) |
| 4 to <6 | 25,694 (93.1) | 1,896 (6.9) | 304 (87.1) | 45 (12.9) |
| ≥6 | 21,088 (93.3) | 1,503 (6.7) | 212 (89.7) | 24 (10.3) |
| *Educational attainment* | | | | |
| High school or less | 25,985 (92.4) | 2,146 (7.6) | 319 (88.1) | 43 (11.9) |
| Junior college | 33,068 (93.0) | 2,492 (7.0) | 353 (86.9) | 53 (13.1) |
| University or higher | 17,770 (93.4) | 1,246 (6.6) | 215 (87.7) | 30 (12.3) |
| *Smoking habit* | | | | |
| Never | 46,530 (93.3) | 3,360 (6.7) | 516 (87.5) | 74 (12.5) |
| Stopped | 27,424 (92.5) | 2,220 (7.5) | 331 (88.9) | 41 (11.1) |
| Smoking | 2,869 (90.4) | 304 (9.6) | 40 (78.2) | 11 (21.8) |
| *Alcohol intake* | | | | |
| Never | 26,900 (93.4) | 1,911 (6.6) | 293 (85.3) | 51 (14.7) |
| Stopped | 42,141 (92.6) | 3,380 (7.4) | 520 (87.8) | 72 (12.2) |
| Drinking | 7,782 (92.9) | 593 (7.1) | 74 (95.9) | 3 (4.1) |
| *Congenital diseases* | | | | |
| Absence | 69,913 (93.0) | 5,224 (7.0) | 753 (87.8) | 105 (12.2) |
| Presence | 6,910 (91.3) | 660 (8.7) | 134 (86.4) | 21 (13.6) |

ASD = autism spectrum disorders; IQR = interquartile range; PRB = parent-reported bruxism; SD = standard deviation.

**Table 2. Association between prevalence of PRB and ASD.**

|  | Control | ASD |  |
|---|---|---|---|
| **Presence, n (%)** | **5,884 (7.1)** | **126 (12.4)** | **_p_-value** |
| Crude | Ref | 1.86 (1.53–2.26) | <0.001 |
| Model 1[a] |  | 1.76 (1.45–2.15) | <0.001 |
| Model 2[b] |  | 1.61 (1.32–1.96) | <0.001 |
| Model 3[c] |  | 1.59 (1.31–1.94) | <0.001 |

ASD = autism spectrum disorders; PRB = parent-reported bruxism.

Odds ratio (95% confidence interval) (all such values).

[a]Adjusted for maternal age and infant's sex.

[b]Additionally adjusted for maternal factors (educational attainment, smoking, and drinking habits), household income, and presence of sibling(s) in infants in Model 1.

[c]Additionally adjusted for the infant's sleep duration at 1 month of age in Model 2.

mean with standard deviation (SD), respectively. Categorical variables were presented as numbers and percentiles. All statistical analyses were conducted using the IBM SPSS Statistics software (version 24.0; IBM Corp., Armonk, NY, USA), and _p_-values less than 0.05 were considered significant.

For analyzing the association of PRB prevalence with that of ASD in children, binomial logistic regression analysis involving the potential covariates obtained using the simultaneous method was performed and the odds ratios (ORs) for PRB were estimated; the control group was used as the reference. For crude or adjusted analyses using the aforementioned covariates, Model 1 was analyzed after adjusting for maternal age at delivery and sex of the child. The variables included in Models 1 and 2 included maternal factors (educational attainment, smoking, and drinking habits), infant factors (presence of siblings and prevalence of congenital diseases), and household income. Additionally, Model 3 included the infants' sleep duration at 1 month of age as a continuous variable (Table 2). ORs and 95% confidence intervals (95% CIs) were calculated for PRB. After stratification by the prevalence of ASD, patient subgroup analysis of the association between PRB and infant sleep duration, which was used both as the continuous and ordinal variable, was performed. When used as an ordinal variable, the group with the shortest sleep duration was designated the reference group.

## Results

The baseline characteristics of the participants according to the prevalence of ASD and PRB (at both 2 and 4 years after delivery) are presented in Table 1. ASD was estimated to be prevalent in 1,014 children (1.2%). Of these, 383 participants (0.5%) were diagnosed 3 years after delivery. Additionally, 13,924 (16.6%) and 19,022 (22.7%) participants presented with bruxism, as reported by caregivers, at 2 and 4 years of age, respectively. PRB was observed in 5,884 (7.1%) and 126 (12.4%) participants in the control and ASD groups, respectively. Notably, 3,470 (8.1%) and 2,540 (6.2%) were males and females with PRB, respectively. Furthermore, the mean sleep duration of participants with PRB was significantly shorter than that of the controls, particularly at 1 month, but not at other months after delivery. As shown in S1 Table, the highest occurrence of PRB was found in the group with the shortest sleep duration among all participants until 6 months after delivery, but not at later time points.

The crude and adjusted ORs of ASD for PRB were calculated using multivariate logistic regression analysis (Table 2). The OR (95% CI) for prevalence of PRB in the adjusted model for all covariates increased with the prevalence of ASD (1.59 [1.31–1.94]). Notably, in an

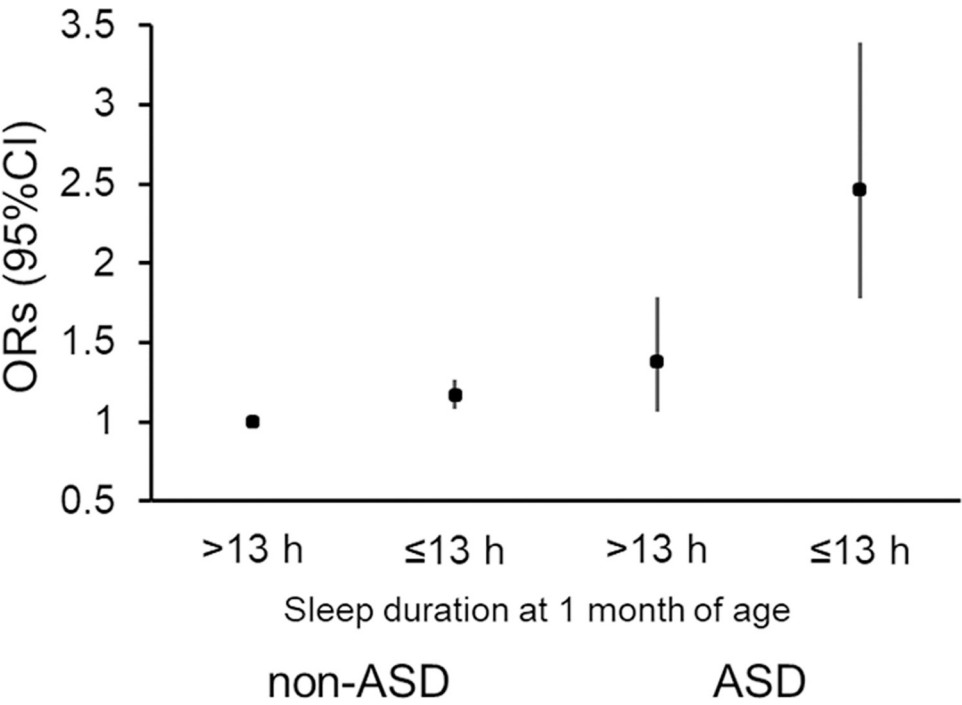

**Fig 2. Interactive impact of ASD prevalence and short sleep duration on the occurrence of PRB at 1 month of age.**

adjusted model for all covariates, the OR (95% CI) for the prevalence of PRB was 0.98 (0.97–0.99) for per hour increase in sleep duration in children aged 1 month.

Next, we focused on the association between children's sleep duration in early childhood and the prevalence of PRB in participants with ASD, stratified by participants with prevalence. The participants' baseline association of sleep duration at each time point with the prevalence of ASD and PRB is presented in S1 Table. In participants of the shortest sleep duration group (used as the reference), the crude and adjusted ORs for the prevalence of PRB decreased consistently with longer sleep duration at 1 month of age regardless of the prevalence of ASD (S2 Table). The ORs (95% CIs) for the prevalence of PRB for per hour increase in sleep duration in the control and ASD groups were 0.98 (0.97–0.99) and 0.95 (0.90–0.99), respectively. However, the lower ORs of longer sleep duration for the prevalence of PRB were not observed after 6 months in participants with ASD (S2 Table).

Furthermore, we analyzed the interaction between ASD prevalence and short sleep duration at 1 month of age ($\leq$13) with the prevalence of PRB. Compared with that in the reference group (participants having normal sleep duration [>13 h] in the control group), short sleep duration (1.17 [1.10–1.25], p < 0.001) and prevalence of ASD (1.38 [1.08–1.78], p = 0.012) were associated with increased ORs (95% CIs) for the prevalence of PRB in the adjusted model for all covariates. The OR (95% CIs) for the prevalence of PRB in ASD participants having short sleep duration was 2.46 (1.80–3.38, p < 0.001) (Fig 2). Notably, a significant interaction between ASD and short sleep duration was observed (1.52 [1.02–2.23], p < 0.05).

## Discussion

In this study, we analyzed the data obtained from a large-scale nationwide birth cohort in Japan. The key findings were that shorter sleep duration in the neonatal period was additively associated with developing PRB in children with ASD, with an approximately 1.6-fold higher

risk of PRB prevalence, than children with typical development. As poor oral health is common among people with ASD, a better understanding of the relationship between bruxism prevalence and sleep issues would help in maintaining the oral health of such individuals.

Shorter sleep duration in the neonatal period remarkably impacts the occurrence of PRB in children regardless of the prevalence of ASD, although the association was more significant in children with ASD. The frequency of sleep disturbances in individuals with both ASD and PRB prevalence has been reported previously [10, 18, 23, 35]. In particular, sleep-onset problems associated with ASD are common in infants [19]. However, studies investigating the mechanism underlying the etiopathology of predominant bruxism in individuals with ASD and its involvement with sleep issues are limited [10, 36]. The mean difference in sleep duration between children aged 1 month with and without prevalence of PRB was higher in those with ASD (1.1 h vs. 0.4 h in the control group). Comparison of the effects between participants with and without ASD showed that the short sleep duration at 1 month of age (≤13 h) was dominantly associated with high prevalence of PRB in children with ASD (Fig 2). Sleep issues increase the risk of bruxism prevalence; however, the evidence provided is insufficient [10, 18]. Brainstem function develops remarkably in newborns and contributes significantly to sleep development in infants [37], as well as to ASD aggravation [38]. Previous studies have outlined the primary targets that potentially induce involuntary bruxism [1, 39]. However, further studies focusing on brain development and behavioral sleep interventions in early childhood are warranted to determine the mechanisms underlying the behavioral development of excessive bruxism in children with ASD, specifically regarding sleep issues during the neonatal period.

A systematic review of case-controlled trials [10] revealed that the prevalence of PRB was consistently high in children with ASD. However, the increased risk in our study (approximately 1.6-fold) was relatively lower than that in a previous systematic review (approximately 4-fold of the control) [10]. In addition to the difference in study designs between case-control and cohort studies, the participants in whom the onset of bruxism in early childhood was investigated were younger than those included in previous studies owing to the design of the birth cohort study. Treatment with ASD medications is generally started at around the preschool age, some of which potentially aggravate bruxism [7, 11]. Additionally, one-third to half of ASD cases were reportedly identified after 6 years in participants with undiagnosed ASD [40]. The proportion of children with ASD in this study was not low [15], although potential patients with ASD were included in the control group. Therefore, the risk of bruxism is expected to be high in mature participants; however, further investigations are required.

A child's sex can directly impact both early-stage neurodevelopment and the prevalence of ASD [14, 15]. Concerning the impact of sex differences on the results, the prevalence of PRB in male participants was higher than that in females (8.1% vs. 6.2% in females), and the OR (95% CI) in males was 1.32 (1.25–1.39, p < 0.001) in the adjusted model for all variables, including the prevalence of ASD, a male-dominant disorder. Despite the controversy regarding the sex-based differences in the prevalence of bruxism owing to the different diagnostic criteria and study designs used in previous studies, the prevalence of bruxism in males, as shown in this study, is consistent with that reported by others [41, 42]. In conclusion, regardless of sex-specific differences, excessive bruxism observed in patients with ASD increases the risk of various oral health problems, including periodontal diseases, malocclusion, and tooth wear [18, 43–45]. Thus, based on the observations regarding the deteriorated oral development and hygiene [12, 13]. our findings can provide valuable information for preventing negative health events in children with ASD.

Our study has several advantages and limitations. The JECS dataset used in our study was obtained from a nationwide Japanese survey that included almost half of all the infants born in several regions from 2011 to 2014 [24, 26, 30]. Thus, the findings, mostly based on the

common Japanese population, allowed us to compare the behavioral onset of PRB in children with ASD and its association with the infants' sleep duration with abundant control participants. However, owing to the collection methods used, limitations associated with this dataset exist, particularly those regarding insufficient information on the type and frequency of bruxism. Since the questionnaire did not include a query regarding the diagnostic validity and classification of bruxism, the PRB included all types and severities of bruxism (e.g., grinding and clenching during sleeping and wakefulness) in this study. In addition, Ishimaru and his colleagues have recently suggested a lower concordance between self-reported and device-detected bruxism (during both sleeping and waking) at an interindividual level [46, 47]. Although unfavorable in early childhood, direct measurements, such as those obtained using electromyography for estimating sleep bruxism, can provide more accurate data during sleeping and waking.

## Conclusion

In conclusion, shorter sleep duration in the neonatal period potentially contributes to an increased risk of developing bruxism in children with ASD. Further studies with accurate estimation of bruxism in children with ASD are warranted.

## Supporting information

**S1 Text. STROBE statement.**
(DOCX)

**S1 Table. Baseline association of sleep duration with the prevalence of PRB in 83,720 children.**
(DOCX)

**S2 Table. Association of sleep duration with the prevalence of PRB in infants with or without ASD.**
(DOCX)

## Acknowledgments

We thank all the JECS participants and the JECS staff members for conducting the procedure and helping with the data analysis.

Members of the JECS Group as of 2023: Michihiro Kamijima (principal investigator, Nagoya City University, Nagoya, Japan, kamijima@med.nagoya-cu.ac.jp), Shin Yamazaki (National Institute for Environmental Studies, Tsukuba, Japan), Yukihiro Ohya (National Center for Child Health and Development, Tokyo, Japan), Reiko Kishi (Hokkaido University, Sapporo, Japan), Nobuo Yaegashi (Tohoku University, Sendai, Japan), Koichi Hashimoto (Fukushima Medical University, Fukushima, Japan), Chisato Mori (Chiba University, Chiba, Japan), Shuichi Ito (Yokohama City University, Yokohama, Japan), Zentaro Yamagata (University of Yamanashi, Chuo, Japan), Hidekuni Inadera (University of Toyama, Toyama, Japan), Takeo Nakayama (Kyoto University, Kyoto, Japan), Tomotaka Sobue (Osaka University, Suita, Japan), Masayuki Shima (Hyogo Medical University, Nishinomiya, Japan), Seiji Kageyama (Tottori University, Yonago, Japan), Narufumi Suganuma (Kochi University, Nankoku, Japan), Shoichi Ohga (Kyushu University, Fukuoka, Japan), and Takahiko Katoh (Kumamoto University, Kumamoto, Japan).

The findings and conclusions of this article are solely the responsibility of the authors and do not represent the official views of the Japanese government. We also thank Editage (www. editage.com) for English language editing.

## Author Contributions

**Conceptualization:** Masahiro Tsuchiya, Haruki Momma.

**Data curation:** Masahiro Tsuchiya, Haruki Momma.

**Formal analysis:** Masahiro Tsuchiya, Shinobu Tsuchiya, Haruki Momma.

**Funding acquisition:** Nobuo Yaegashi, Takahiro Arima, Chiharu Ota.

**Methodology:** Masahiro Tsuchiya, Haruki Momma.

**Resources:** Nobuo Yaegashi, Takahiro Arima, Chiharu Ota.

**Supervision:** Ryoichi Nagatomi, Nobuo Yaegashi, Chiharu Ota, Kaoru Igarashi.

**Writing – original draft:** Masahiro Tsuchiya, Shinobu Tsuchiya.

**Writing – review & editing:** Shinobu Tsuchiya, Haruki Momma, Ryoichi Nagatomi, Kaoru Igarashi.

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
