## [Decision Letter · Decision Letter 0]

5 Jun 2024

PONE-D-24-13310Behavioral occurrence of bruxism in children with autism spectrum disorder:the Japan Environment and Children's Study (JECS)PLOS ONE

Dear Dr. Tsuchiya,

Thank you for submitting your manuscript to PLOS ONE. After careful consideration, we feel that it has merit but does not fully meet PLOS ONE’s publication criteria as it currently stands. Therefore, we invite you to submit a revised version of the manuscript that addresses the points raised during the review process.

Your manuscript has been reviewed by two expert reviewers. I would like to inform you that both reviewers have not recommended publication of this manuscript in its present form. Although the manuscript presents interesting new findings, it also contains some drawbacks, as clearly described in their comments.

I would like to encourage you to revise the manuscript extensively according to their comments. You can find their comments at the end of this e-mail.

We look forward to receiving your revised manuscript.

Kind regards,

Ayako Mochizuki

Academic Editor

PLOS ONE

2. Please amend either the title on the online submission form (via Edit Submission) or the title in the manuscript so that they are identical.

3. One of the noted authors is a group or consortium [Japan Environment and Children’s Study Group (JECS)]. In addition to naming the author group, please list the individual authors and affiliations within this group in the acknowledgments section of your manuscript. Please also indicate clearly a lead author for this group along with a contact email address.

Additional Editor Comments:

Your paper has been reviewed. The comments of the reviewers are included at the bottom of this letter.

The reviewers have recommended major revisions to your manuscript. Therefore, I invite you to revise and resubmit your manuscript as fast as possible.

Please carefully address the issues raised in the comments.

Kind regards,

Ayako Mochizuki

Academic Editor

PLOS ONE

Reviewers' comments:

Reviewer's Responses to Questions

**Comments to the Author**

1. Is the manuscript technically sound, and do the data support the conclusions?

Reviewer #1: Yes

Reviewer #2: Partly

2. Has the statistical analysis been performed appropriately and rigorously? 

Reviewer #1: Yes

Reviewer #2: Yes

3. Have the authors made all data underlying the findings in their manuscript fully available?

Reviewer #1: Yes

Reviewer #2: Yes

4. Is the manuscript presented in an intelligible fashion and written in standard English?

Reviewer #1: Yes

Reviewer #2: Yes

5. Review Comments to the Author

Reviewer #1: In this study, a dataset (n = 83,720) obtained from the Japan Environment and Children's Study (JECS), a nationwide birth cohort study was subjected to multiple imputation for verifying an increased prevalence rate of bruxism behavior in children with ASD. Additionally, the association between habitual bruxism in participants with ASD and sleep duration in infants on bruxism was examined using logistic regression analysis with adjustments for several maternal and child-related variables. The prevalence of ASD and habitual bruxism was 1.2% and 7.2%,

respectively. The odds ratio of the increased risk of bruxism in individuals with ASD (95% confidence interval) was 1.59 (1.31–1.94) after covariate adjustments. Furthermore, a longer sleep duration in the neonatal period was significantly associated with a decreased risk of habitual bruxism in participants with ASD. The results of this study

show that habitual bruxism is highly prevalent among children with ASD and is associated with sleep duration, particularly in the neonatal stage. The results of this study are interesting, and this article is well written. However, the following minor concerns should be addressed.

1) Emerging evidences suggest that dysbiosis of gut microbiota plays a role in ASD, and that oral microbiota may in part be regulated by gut microbiota. Please discuss a potential role of dysbiosis in oral microbiota and subsequent poor oral care in bruxism of ASD children.

2) Table 1: Prevalence of ASD in boy is higher than that of girl. is there a significant gender difference in bruxism in this study?

Overall, this study was conducted well, and this article is interesting.

Reviewer #2: [General comments]

This study attempted to examine the increased prevalence of bruxism in children with ASD, and the relationship between the prevalence of bruxism and sleep duration in infants. The objective of the study was vague, the expression of novelty in the introduction and discussion is unclear, and the differences with the results of the previous papers are unclear. Furthermore, the definition of bruxism was not explained sufficiently. Bruxism was assessed solely through a questionnaire, but the diagnostic validity of this is questionable.

Without sufficient explanation and improvements on these critical points, it is difficult to recommend this paper for publication in PLOS ONE.

[Detailed concerns]

Introduction

#1

The authors should clarify what had not been clarified in the previous researches and mention specifically which part of the characteristics of the relation between bruxism and ASD was tried to clarify in this research. More clear statement on objectives is needed. It is probably better method to present a specific hypothesis.

#2

Is the bruxism in ASD patients classified as secondary bruxism (symptomatic bruxism)?

It should be stated how the previous studies dealt with and how the authors had recognized for conducting this study.

#3

In the introduction part, some comments on drug-induced factors on ASD and bruxism should also be described.

#4 It should also be made clear whether the bruxism objected in this study is sleep bruxism and/or awake bruxism.

Methods

#5

A clear definition of "bruxism" should be presented. Further, the definition of "habitual bruxism" should be made clear in terms of definitions in previous studies, definitions at international conferences etc. In addition, the local definition of "bruxism" within this paper and the reason for adopt the definition should also be stated.

#6

It is necessary to provide a detailed explanation of the method for assessing bruxism. The contents of the reference studies should also be explained. The validity of diagnosing by interview alone should be discussed in the method part and/or the discussion part. This might had better be added as a limitation in the discussion part.

Results

#7

The following items should be written in more understandable manner: 1. The relationship between sleep duration and the prevalence of bruxism, 2. The relationship between ASD and the prevalence of bruxism, and 3. Whether the influence of sleep duration is greater in ASD patients than in controls.

#8

P15L218

It is questionable whether the following sentences can explain the characteristic that bruxism of individuals with ASD are more susceptible to the influences of sleep duration than controls.

“The ORs (95%CIs) for habitual bruxism per hour increase of sleep duration in the control and ASD groups were 0.98 (0.97–0.99) and 0.95 (0.90–0.99), respectively. However, the lower ORs for habitual bruxism were additive with longer sleep duration at 6 months of age in the control group but not in participants with ASD. Additionally, the association between sleep duration at later time points, from 12 to 36 months of age, and habitual bruxism was not significant (S2 Table).”

#9

The following two comments about the previous studies seemed evoke doubt on the novelty of this study. The authors should provide more detailed explanation how the findings of this study relate to those of previous studies.

P16L232 “the crude and adjusted ORs for habitual bruxism behavior consistently decreased with longer sleep duration at 1 month of age, similar to our latest report [10].”

P18L250 “We found that the occurrence of habitual bruxism was higher in children with ASD than in those with typical development, which is consistent with the findings of a systematic review of case-controlled trials by [17]”

#10

P19L277　ASD-related bruxism

Please explain if ASD-related bruxism is an official name of classification of bruxism? Does it refer to one of secondary bruxism?

Conclusion

#11

The novelty in the following simple conclusions is difficult for readers to understand. More adequate expression of explanation is considered to be needed as conclusion.

P21L301 “The risk of developing habitual bruxism is higher in children with ASD than those without the disorder and is associated with shorter sleep duration in the neonatal period.”

6. PLOS authors have the option to publish the peer review history of their article (what does this mean?). If published, this will include your full peer review and any attached files.

Reviewer #1: No

Reviewer #2: No

---

## [Author Response · Author response to Decision Letter 0]

16 Jul 2024

PONE-D-24-13310

Title: Behavioral occurrence of bruxism in children with autism spectrum disorder:the Japan Environment and Children's Study (JECS)

Authors: Masahiro Tsuchiya, Shinobu Tsuchiya, Haruki Momma, Ryoichi Nagatomi, Nobuo Yaegashi, Takahiro Arima, Chiharu Ota, Kaoru Igarashi, and the Japan Environment and Children’s Study Group.

Dear Dr. Ayako Mochizuki, DDS, PhD.

Academic Editor in PLOS ONE

We wish to thank the editor and reviewers for carefully reading our manuscript and providing detailed comments and suggestions, which have helped improve the manuscript. We have considered them and have revised the manuscript accordingly. Our changes are shown in red. Point-by-point answers to the comments of the editor and reviewers are in bold below:

Reviewer #1: In this study, a dataset (n = 83,720) obtained from the Japan Environment and Children's Study (JECS), a nationwide birth cohort study was subjected to multiple imputation for verifying an increased prevalence rate of bruxism behavior in children with ASD. Additionally, the association between habitual bruxism in participants with ASD and sleep duration in infants on bruxism was examined using logistic regression analysis with adjustments for several maternal and child-related variables. The prevalence of ASD and habitual bruxism was 1.2% and 7.2%, respectively. The odds ratio of the increased risk of bruxism in individuals with ASD (95% confidence interval) was 1.59 (1.31–1.94) after covariate adjustments. Furthermore, a longer sleep duration in the neonatal period was significantly associated with a decreased risk of habitual bruxism in participants with ASD. The results of this study show that habitual bruxism is highly prevalent among children with ASD and is associated with sleep duration, particularly in the neonatal stage. The results of this study are interesting, and this article is well written. However, the following minor concerns should be addressed.

We are grateful for your careful review of our manuscript and useful comments, which helped us address the concerns raised with the appropriate corrections and revisions. We have considered your suggestions and revised the manuscript accordingly.

1) Emerging evidences suggest that dysbiosis of gut microbiota plays a role in ASD, and that oral microbiota may in part be regulated by gut microbiota. Please discuss a potential role of dysbiosis in oral microbiota and subsequent poor oral care in bruxism of ASD children.

A hypothesis of dysbiosis in children with ASD should be interesting for readers. Following the suggestion from reviewer #1, we added the relevant points to the Discussion section (P19L303–P21L307).

2) Table 1: Prevalence of ASD in boy is higher than that of girl. is there a significant gender difference in bruxism in this study?

In the JECS dataset, the prevalence of bruxism behavior in male participants was higher than in females (8.1% vs 6.2% in females), and the OR (95% CI) of bruxism behavior in males was 1.32 (1.25–1.39, p <0.001) in the adjusted model for all covariates. Because the results definitively depend upon the diagnostic criteria and study designs, the sex-based difference in bruxism behavior remains controversial. Several reports show a dominant prevalence of bruxism behavior in males (Lam MH, et al., Sleep Med. 2011;12:641-5; Insana SP, et al, Sleep Med. 2013;14:183-8; de Almeida AB, et al., Int J Environ Res Public Health. 2022;19(13):7823), in female (Serra-Negra JM, et al., Eur Arch Paediatr Dent. 2010;11(4):192–195.; Seraj B, et al., Iran J Pediatr. 2010 Jun; 20(2): 174–180.; Alves CL, Sleep Sci. 2019; 12(3): 185–189.), and no differences (Fonseca CM, et al., Sleep Breath. 2011;15(2):215–220.). Of these studies, Lam et al. (Sleep Med. 2011;12:641-5), based on a multivariate logistic regression analysis, indicated an increased OR for bruxism behavior in males (OR [95% CI] = 1.69 [1.37–2.10], p <0.001).

Thus, the above descriptions have been summarized in the Results and Discussion sections to ensure clarity in sex-based differences in bruxism behavior (P13L213–214; P19L292–301).

Overall, this study was conducted well, and this article is interesting.

We thank Reviewer #1 for the careful review and structural suggestions, which have helped us improve the manuscript.

Reviewer #2: [General comments]

This study attempted to examine the increased prevalence of bruxism in children with ASD, and the relationship between the prevalence of bruxism and sleep duration in infants. The objective of the study was vague, the expression of novelty in the introduction and discussion is unclear, and the differences with the results of the previous papers are unclear. Furthermore, the definition of bruxism was not explained sufficiently. Bruxism was assessed solely through a questionnaire, but the diagnostic validity of this is questionable. Without sufficient explanation and improvements on these critical points, it is difficult to recommend this paper for publication in PLOS ONE.

We thank Reviewer #2 for the careful review, detailed comments, and structural suggestions, which have helped us to improve the quality of the manuscript. We have accepted all of the suggestions for revision and thoroughly made appropriate changes throughout the manuscript, to ensure clarity for the journal’s readers. Specifically, the study’s purpose (P3L40–42 in abstract; P5L80–P6L87), the novel findings in our research (P3L45–L49 in abstract; P17L253–258; P21L325–326), the definition of bruxism (P4L51–53; P7L120–P8L122), and the study’s limitation related to the diagnostic validity of bruxism by the questionnaire (P20L313–P21L322) have been described under appropriate sections of the manuscript. We hope that the revised manuscript is deemed suitable for publication in PLOS ONE.

Introduction

#1

The authors should clarify what had not been clarified in the previous researches and mention specifically which part of the characteristics of the relation between bruxism and ASD was tried to clarify in this research. More clear statement on objectives is needed. It is probably better method to present a specific hypothesis.

We appreciate your structural comments. Severe bruxism is commonly observed in children with ASD and is considered a major oral health concern damaging orofacial tissues such as teeth, periodontium, and temporomandibular joints. A recent systematic review by Granja (Spec Care Dentist. 2022;42(5):476-485) reported that bruxism’s prevalence is approximately four times higher among individuals with ASD (P4L61–62). Since the association between a higher occurrence of bruxism and ASD has been acknowledged, we focused on the additive interaction between infant sleep duration and the occurrence of bruxism among children with ASD as a specific hypothesis in this manuscript (P5L84 – P6L87). 

#2

Is the bruxism in ASD patients classified as secondary bruxism (symptomatic bruxism)?

It should be stated how the previous studies dealt with and how the authors had recognized for conducting this study.

As pointed out by Reviewer #2, bruxism is distinguished as primary or secondary based on the association with other underlying diseases (Bulanda S, et al. Int J Environ Res Public Health. 2021;18(18):9544). Thus, bruxism behavior observed in patients with ASD is categorized as secondary bruxism (symptomatic bruxism). In agreement with your suggestion, we stated the detailed definition of bruxism involving the classification in the Introduction (P4L58–L61).

#3

In the introduction part, some comments on drug-induced factors on ASD and bruxism should also be described.

We appreciate your pertinent comment. de Baat et al. summarized ASD medications potentially aggravating bruxism behavior (J Oral Rehabil. 2021;48:343-354.). According to your suggestion, we added this information in the Introduction (P4L62–L65). The characteristics of participants, mostly young, before the start of ASD medication, have been described in the section of the study limitation (P18L282–P19L291).

#4 It should also be made clear whether the bruxism objected in this study is sleep bruxism and/or awake bruxism.

We are grateful for your careful review of our manuscript. The bruxism behavior is distinguished into two circadian symptoms: sleep and awake (P4L51–L53; P20L316–L320), as pointed out by Reviewer #2. Regarding the main study limitation, the dataset did not include enough information on bruxism behavior observed during sleep/awake status. Although further studies with detailed estimation of bruxism behavior in children with ASD are warranted, it should be noted that Winocur and his colleagues previously observed a higher concordance of both sleep and awake bruxism at an interindividual level (Front Neurol 2019;10:443.). We added the above including the reference under limitations in the revised manuscript (P20L313–P21L322).

Methods

#5

A clear definition of "bruxism" should be presented. Further, the definition of "habitual bruxism" should be made clear in terms of definitions in previous studies, definitions at international conferences etc. In addition, the local definition of "bruxism" within this paper and the reason for adopt the definition should also be stated.

We appreciate your valuable comment to improve the manuscript. First, according to the international consensus on bruxism definition (Lobbezoo F, Ahlberg J, Raphael KG, et al. International consensus on the assessment of bruxism: Report of a work in progress. J Oral Rehabil. 2018;45:837-844.), the definition in the Abstract and Introduction has been corrected (P3L34-35 in Abstract; P4L51–53). 

We changed the term “habitual bruxism” to “bruxism behavior” throughout the manuscript for ease of understanding of our findings for readers and reviewers. Additionally, in terms of the definition for the local definition of “bruxism behavior” in this study, considering that consistent bruxism behavior increases a risk for dental tissue damage, participants with “bruxism behavior” observed at both time points (2 and 4 years of age) were categorized into the bruxism group throughout the manuscript. To help readers understand better, we have added a clear categorization explaining the above in the Methods section (P7L117–P8L124). 

#6

It is necessary to provide a detailed explanation of the method for assessing bruxism. The contents of the reference studies should also be explained. The validity of diagnosing by interview alone should be discussed in the method part and/or the discussion part. This might had better be added as a limitation in the discussion part.

We are grateful for your careful reading of the manuscript and structural suggestions. Since the data collection methods in this study did not include a query about the diagnostic validity and classification of bruxism behavior, the assessment of a child's bruxism behavior only using the parental-report questionnaire should be considered as a major study limitation. For example, a direct measure, such as electromyography for estimating sleep bruxism would provide more accurate data in sleep-wake cycles. In contrast, it would not be favorable for a large-scale cohort study in early childhood like in this study. Conclusively, further studies with detailed estimation of bruxism behavior in children with ASD are warranted. In agreement with your point, we added and revised the sentences explaining the above under limitations in the Discussion section (P20L313–P21L322).

Results

#7

The following items should be written in more understandable manner: 1. The relationship between sleep duration and the prevalence of bruxism, 2. The relationship between ASD and the prevalence of bruxism, and 3. Whether the influence of sleep duration is greater in ASD patients than in controls.

Thank you for your valuable comment. We revised the Result section according to your comment above.

 The OR (95% CIs) for “1. the relationship between sleep duration and the prevalence of bruxism” was 0.98 (0.97–0.99) per hour increase. However, since it is consistent with our previous study (Tsuchiya et al., Sleep Med. 2022:100:71-78.), it is stated on P15L223–224. For “2. The relationship between ASD and the prevalence of bruxism,” the OR (95% CIs) was 1.59 [1.31–1.94] as described on P15L221-223 and Table 2.

 “3. Whether the influence of sleep duration is greater in ASD patients than in controls” was additionally examined and indicated in Figure 2. Regarding the interaction between ASD prevalence and the shortest sleep duration at 1 month of age (≤13) on bruxism behavior, short sleep duration (1.17 [1.10–1.25], p <0.001) and prevalence of ASD (1.38 [1.08–1.78], p = 0.012) were associated with increased ORs (95% CIs) for bruxism behavior in the adjusted model for all covariates. Further, the OR (95% CIs) for bruxism behavior in ASD participants having short sleep duration was 2.46 (1.80–3.38, p <0.001) (Figure 2). Furthermore, a significant interaction between ASD and short sleep duration was observed (1.52 [1.02–2.23], p <0.05).” Thus, in agreement with your suggestion, the above description has been newly included in the Result section (P16L238– 246).

#8

P15L218

It is questionable whether the following sentences can explain the characteristic that bruxism of individuals with ASD are more susceptible to the influences of sleep duration than controls.

“The ORs (95%CIs) for habitual bruxism per hour increase of sleep duration in the control and ASD groups were 0.98 (0.97–0.99) and 0.95 (0.90–0.99), respectively. However, the lower ORs for habitual bruxism were additive with longer sleep duration at 6 months of age in the control group but not in participants with ASD. Additionally, the association between sleep duration at later time points, from 12 to 36 months of age, and habitual bruxism was not significant (S2 Table).”

We thank Reviewer #2 for the structural suggestions, which have helped us improve the manuscript. As you pointed out, these sentences can cause misunderstanding for readers. After analyzing the interaction between infants’ sleep duration and ASD prevalence, we revised the description and added Figure 2 as described in our response to comment #7 (P16L238– 246).

#9

The following two comments about the previous studies seemed evoke doubt on the novelty of this study. The authors should provide more detailed explanation how the findings of this study relate to those of previous studies.

P16L232 “the crude and adjusted ORs for habitual bruxism behavior consistently decreased with longer sleep duration at 1 month of age, similar to our latest report [10].”

P18L250 “We found that the occurrence of habitual bruxism was higher in children with ASD than in those with typical development, which is consistent with the findings of a systematic review of case-controlled trials by [17]”

We understand the reviewer’s argument that the sentences were vague. Thus, we have rewritten them (P16L233–237; P17L253–256).

#10

P19L277　ASD-related bruxism

Please explain if ASD-related bruxism is an official name of classification of bruxism? Does it refer to one of secondary bruxism?

Bruxism observed in patients with ASD is categorized as secondary as described in response to #2. In accordance with the suggestions, the term “ASD-related bruxism” has been revised to “bruxism behavior in patients with ASD” (P4L61–65; P19L301).

Conclusion

#11

The novelty in the following simple conclusions is difficult for readers to understand. More adequate expression of explanation is considered to be needed as conclusion.

P21L301 “The risk of developing habitual bruxism is higher in children with ASD than those without the disorder and is associated with shorter sleep duration in the neonatal period.”

The Conclusion has been revised as per your suggestions (P21L325–326).

We again thank the reviewers for providing insightful comments.

---

## [Decision Letter · Decision Letter 1]

15 Aug 2024

PONE-D-24-13310R1Behavioral occurrence of bruxism in children with autism spectrum disorder:the Japan Environment and Children's StudyPLOS ONE

Dear Dr. Tsuchiya,

Thank you for submitting your manuscript to PLOS ONE. After careful consideration, we feel that it has merit but does not fully meet PLOS ONE’s publication criteria as it currently stands. Therefore, we invite you to submit a revised version of the manuscript that addresses the points raised during the review process.

We look forward to receiving your revised manuscript.

Kind regards,

Ayako Mochizuki

Academic Editor

PLOS ONE

**Additional Editor Comments:**

Your paper has been reviewed. The comments of the reviewers are included at the bottom of this letter.

The reviewers have recommended major revisions to your manuscript. Therefore, I invite you to revise and resubmit your manuscript as fast as possible.

Please carefully address the issues raised in the comments.

Kind regards,

Ayako Mochizuki

Academic Editor

PLOS ONE

Reviewers' comments:

Reviewer's Responses to Questions

**Comments to the Author**

1. If the authors have adequately addressed your comments raised in a previous round of review and you feel that this manuscript is now acceptable for publication, you may indicate that here to bypass the “Comments to the Author” section, enter your conflict of interest statement in the “Confidential to Editor” section, and submit your "Accept" recommendation.

Reviewer #1: All comments have been addressed

Reviewer #2: (No Response)

2. Is the manuscript technically sound, and do the data support the conclusions?

Reviewer #1: Yes

Reviewer #2: No

3. Has the statistical analysis been performed appropriately and rigorously? 

Reviewer #1: Yes

Reviewer #2: Yes

4. Have the authors made all data underlying the findings in their manuscript fully available?

Reviewer #1: Yes

Reviewer #2: Yes

5. Is the manuscript presented in an intelligible fashion and written in standard English?

Reviewer #1: Yes

Reviewer #2: Yes

6. Review Comments to the Author

Reviewer #1: My all comments have been addressed. The authors addressed all comments. I have no additional comments.

Reviewer #2: [General comment]

Although the issues raised by Reviewers have been somewhat improved, a more appropriate explanation based on deeper understanding of bruxism is necessary.

[Detailed concerns]

1）”bruxism behavior”

------------The authors are responsible for accurately defining the extent and characteristics of the study participants.

The subjects in this study were bruxers that were determined by whether or not they were aware of their bruxism based on report by their parents.

More specifically, the subjects in this study were child bruxers whose parents answered "Yes" to the question, "Does your child have bruxism?" at the ages of 2 and 4.

First of all, this should be clearly stated and appropriate group name (i.e., abbreviation) for the subject group specific to this study should be defined.

Bruxism is a diagnostic term, and the expression "bruxism behavior" is inappropriate because it leads readers to imagine a specific type of bruxism and confuses them. Use of "bruxism behavior" cannot be acceptable.

2）The reply from the author ”in terms of the definition for the local definition of “bruxism behavior” in this study, considering that consistent bruxism behavior increases a risk for dental tissue damage, participants with “bruxism behavior” observed at both time points (2 and 4 years of age) were categorized into the bruxism group throughout the manuscript.”

---------- Since basis and meaning of this theory are unclear, the sentences cannot be acceptable.

3）The question "Does your child have Bruxism?"

-----------What does the question "Does your child have Bruxism?" mean?

Does this question refer to grinding, clenching, during sleep, or during awake state, or include all of them?

More information and explanation are needed in the Methods and Discussion parts regarding the meaning of this question wording.

4）The reply from the author ”The bruxism behavior is distinguished into two circadian symptoms: sleep and awake (P4L51–L53; P20L316–L320), as pointed out by Reviewer #2. Regarding the main study limitation, the dataset did not include enough information on bruxism behavior observed during sleep/awake status. Although further studies with detailed estimation of bruxism　behavior in children with ASD are warranted, it should be noted that Winocur and his colleagues previously observed a higher concordance of both sleep and awake bruxism at an interindividual level (Front Neurol 2019;10:443.). We added the above including the reference under limitations in the revised manuscript (P20L313–P21L322).”

------This reply does not address the reviewer's point that asked for clarification regarding the type of bruxism that was the subject of this study.

It should be more clearly stated in the Introduction and Methods parts whether the bruxism studied in this study was sleep bruxism or awake bruxism.

I cannot understand the intention behind suddenly citing only the paper by Winocur's group. The paper by Winocur's group assessed bruxism based only on simple questions. It is already known that questionnaires do not have a high accuracy rate for either SB or AB.

In particular, recent studies have shown that AB cannot be judged by questionnaire based on self-awareness (Ishimaru, J Prosthodont Res. 2024). In addition, a study that evaluated SB with higher reliability using electromyograms instead of questionnaires found no correlation between SB and AB at all (Mikami, J Prosthodont Res. 2024).

This paper should more clearly state its inability to distinguish between SB and AB as its limitations.

7. PLOS authors have the option to publish the peer review history of their article (what does this mean?). If published, this will include your full peer review and any attached files.

Reviewer #1: No

Reviewer #2: No

---

## [Author Response · Author response to Decision Letter 1]

10 Sep 2024

PONE-D-24-13310

Title: Bruxism associated with short sleep duration in children with autism spectrum disorder: the Japan Environment and Children's Study

Authors: Masahiro Tsuchiya, Shinobu Tsuchiya, Haruki Momma, Ryoichi Nagatomi, Nobuo Yaegashi, Takahiro Arima, Chiharu Ota, Kaoru Igarashi, and the Japan Environment and Children’s Study Group.

Dear Dr. Ayako Mochizuki, DDS, PhD.

Academic Editor in PLOS ONE

We thank the editor and reviewers for repeatedly reading our manuscript and providing structural suggestions, which have helped improve the manuscript. We have considered these suggestions and have revised the manuscript accordingly. Our changes in the manuscript are shown in red. The point-by-point answers to the comments of the editor and reviewers are presented in bold below:

Reviewer #1: My all comments have been addressed. The authors addressed all comments. I have no additional comments.

We thank you for the thoughtful review of our manuscript and useful comments, which have helped us address the concerns raised with appropriate corrections and revisions. 

Reviewer #2:

Although the issues raised by Reviewers have been somewhat improved, a more appropriate explanation based on deeper understanding of bruxism is necessary.

We thank you for carefully reviewing our manuscript and providing constructive suggestions, which have helped us in revising it appropriately. We have considered your suggestions related to the inappropriate definition of bruxism observed in the study participants. We have accordingly modified the relevant sentences in the text. 

1）”bruxism behavior”

------------The authors are responsible for accurately defining the extent and characteristics of the study participants.

The subjects in this study were bruxers that were determined by whether or not they were aware of their bruxism based on report by their parents. More specifically, the subjects in this study were child bruxers whose parents answered "Yes" to the question, "Does your child have bruxism?" at the ages of 2 and 4. First of all, this should be clearly stated and appropriate group name (i.e., abbreviation) for the subject group specific to this study should be defined. Bruxism is a diagnostic term, and the expression "bruxism behavior" is inappropriate because it leads readers to imagine a specific type of bruxism and confuses them. Use of "bruxism behavior" cannot be acceptable.

As pointed out, the study participants were defined as those with parent-reported bruxism for readers’ ease of understanding. We believe that a consistent prevalence of parent-reported bruxism will improve research reproducibility when a parent-administered questionnaire is used for examining the prevalence of bruxism. Thus, we have included participants with bruxism observed by the parents at both time points (2 and 4 years of age) in the PRB group and have changed the term “bruxism behavior” to “parent-reported bruxism (PRB)” throughout the manuscript to avoid misunderstanding.

As we have added the above description in the Methods (page 7, line 121–page 8, line 128), we hope that the term, “parent-reported bruxism”, observed in the participants, and its abbreviation, “PRB”, would address your concerns. Please let us know if issues regarding the handling of this term and its abbreviation persist in the revised manuscript such that we can address them suitably.

2） The reply from the author ”in terms of the definition for the local definition of “bruxism behavior” in this study, considering that consistent bruxism behavior increases a risk for dental tissue damage, participants with “bruxism behavior” observed at both time points (2 and 4 years of age) were categorized into the bruxism group throughout the manuscript.”

---------- Since basis and meaning of this theory are unclear, the sentences cannot be acceptable.

We agree with you that the previous theory explaining the categorization method of parent-reported bruxism was not sufficient in terms of the basis and meaning of the theory. In accordance with your suggestion, we have revised the relevant points in the Methods (page 8, line 124–line 126).

3）The question "Does your child have Bruxism?"

-----------What does the question "Does your child have Bruxism?" mean?

Does this question refer to, or include all of them? More information and explanation are needed in the Methods and Discussion parts regarding the meaning of this question wording.

We understand your argument that the question “Does your child have bruxism?” was vague; however, unfortunately, the questionnaire did not include other information on bruxism observed by the parents. Therefore, our findings included all types and severities of bruxism, for example, grinding and clenching both in the and sleep/awake states, as you have pointed out. Thus, per your suggestion, we have provided further information and explanation in the Methods and Discussion (page 7, line 121–line 128; page 20, line 317–line 322).

4）The reply from the author ”The bruxism behavior is distinguished into two circadian symptoms: sleep and awake (P4L51–L53; P20L316–L320), as pointed out by Reviewer #2. Regarding the main study limitation, the dataset did not include enough information on bruxism behavior observed during sleep/awake status. Although further studies with detailed estimation of bruxism behavior in children with ASD are warranted, it should be noted that Winocur and his colleagues previously observed a higher concordance of both sleep and awake bruxism at an interindividual level (Front Neurol 2019;10:443.). We added the above including the reference under limitations in the revised manuscript (P20L313–P21L322).”

----- -This reply does not address the reviewer's point that asked for clarification regarding the type of bruxism that was the subject of this study. 

It should be more clearly stated in the Introduction and Methods parts whether the bruxism studied in this study was sleep bruxism or awake bruxism.

I cannot understand the intention behind suddenly citing only the paper by Winocur's group. The paper by Winocur's group assessed bruxism based only on simple questions. It is already known that questionnaires do not have a high accuracy rate for either SB or AB.

In particular, recent studies have shown that AB cannot be judged by questionnaire based on self-awareness (Ishimaru, J Prosthodont Res. 2024). In addition, a study that evaluated SB with higher reliability using electromyograms instead of questionnaires found no correlation between SB and AB at all (Mikami, J Prosthodont Res. 2024).

This paper should more clearly state its inability to distinguish between SB and AB as its limitations.

We appreciate your suggestions. The type of bruxism (e.g. sleep or awake bruxism) must be clearly noted. However, the question used in the study could not definitively distinguish the bruxism type into sleep and awake states. Additionally, as suggested, the low accuracy rates for assessing sleep/awake bruxism in studies using questionnaires have been mentioned (Ishimaru et al, J Prosthodont Res. 2024; Mikami et al, J Prosthodont Res. 2024). Following your suggestion, we have revised the relevant points in the Discussion, citing the above references (page 20, line 317–page 21, line 324).

We again thank the reviewer #2 for your careful reading of the manuscript and their appropriate opinions. We have considered your suggestions and revised the manuscript accordingly.

---

## [Decision Letter · Decision Letter 2]

17 Oct 2024

Bruxism associated with short sleep duration in children with autism spectrum disorder:the Japan Environment and Children's Study

PONE-D-24-13310R2

Dear Dr. Masahiro Tsuchiya,

We’re pleased to inform you that your manuscript has been judged scientifically suitable for publication and will be formally accepted for publication once it meets all outstanding technical requirements.

Kind regards,

Ayako Mochizuki

Academic Editor

PLOS ONE

Additional Editor Comments (optional):

I am glad to say that reviewers and I are satisfied with your manuscript and have decided that it is appropriate to publish it in PLOS ONE. Congratulations on your excellent work!

Reviewers' comments:

Reviewer's Responses to Questions

**Comments to the Author**

1. If the authors have adequately addressed your comments raised in a previous round of review and you feel that this manuscript is now acceptable for publication, you may indicate that here to bypass the “Comments to the Author” section, enter your conflict of interest statement in the “Confidential to Editor” section, and submit your "Accept" recommendation.

Reviewer #2: (No Response)

Reviewer #3: All comments have been addressed

2. Is the manuscript technically sound, and do the data support the conclusions?

Reviewer #2: (No Response)

Reviewer #3: Yes

3. Has the statistical analysis been performed appropriately and rigorously? 

Reviewer #2: (No Response)

Reviewer #3: Yes

4. Have the authors made all data underlying the findings in their manuscript fully available?

Reviewer #2: (No Response)

Reviewer #3: Yes

5. Is the manuscript presented in an intelligible fashion and written in standard English?

Reviewer #2: (No Response)

Reviewer #3: No

6. Review Comments to the Author

Reviewer #2: (No Response)

Reviewer #3: After reviewing your manuscript, I am pleased to inform you that I found no significant issues. The work is well-structured and contributes meaningfully to the field.

7. PLOS authors have the option to publish the peer review history of their article (what does this mean?). If published, this will include your full peer review and any attached files.

Reviewer #2: No

Reviewer #3: No

---

## [Editor Report · Acceptance letter]

28 Nov 2024

PONE-D-24-13310R2 

PLOS ONE

Dear Dr. Tsuchiya, 

I'm pleased to inform you that your manuscript has been deemed suitable for publication in PLOS ONE. Congratulations! Your manuscript is now being handed over to our production team.

Kind regards, 

on behalf of

Dr. Ayako Mochizuki 

Academic Editor

PLOS ONE